DATA RELEASE

# Annotation of putative circadian rhythm-associated genes in *Diaphorina citri* (Hemiptera: Liviidae)

Max Reynolds[1], Lucas de Oliveira[1], Chad Vosburg[1,2], Thomson Paris[3], Crissy Massimino[1], Jordan Norus[1], Yasmin Ortiz[1], Michelle Espino[1], Nina Davis[1], Ron Masse[1], Alan Neiman[1], Rachel Holcomb[1], Kylie Gervais[1], Melissa Kemp[1], Maria Hoang[1], Teresa D. Shippy[4], Prashant S. Hosmani[5], Mirella Flores-Gonzalez[5], Kirsten Pelz-Stelinski[6], Jawwad A. Qureshi[7,8], Lukas A. Mueller[5], Wayne B. Hunter[9], Joshua B. Benoit[10], Susan J. Brown[4], Tom D'Elia[1] and Surya Saha[5,11,*]

1 Indian River State College, Fort Pierce, FL 34981, USA
2 Department of Plant Pathology and Environmental Microbiology, Pennsylvania State University, University Park, PA 16802, USA
3 Entomology and Nematology Department, University of Florida, North Florida Research and Education Center, Research Road, Quincy 32351, Florida, USA
4 Division of Biology, Kansas State University, Manhattan, KS 66506, USA
5 Boyce Thompson Institute, Ithaca, NY 14853, USA
6 Department of Entomology and Nematology, University of Florida, Lake Alfred, FL 33850, USA
7 Indian River Research and Education Center, University of Florida, IFAS, 2199 South Rock Road, Fort Pierce, FL 34945-3138, USA
8 Southwest Florida Research and Education Center, University of Florida, IFAS, 2685 State Road 29 North, Immokalee, FL 34142, USA
9 USDA-ARS, US Horticultural Research Laboratory, Fort Pierce, FL 34945, USA
10 Department of Biological Sciences, University of Cincinnati, Cincinnati, OH 45221, USA
11 Animal and Comparative Biomedical Sciences, University of Arizona, Tucson, AZ 85721, USA

**Submitted:** 10 October 2021

\* Corresponding author. E-mail: suryasaha@cornell.edu

Preprint submitted at https://doi.org/10.1101/2021.10.09.463768

Included in the series: *Asian citrus psyllid community annotation* (https://doi.org/10.46471/GIGABYTE_SERIES_0001)

## ABSTRACT

The circadian rhythm involves multiple genes that generate an internal molecular clock, allowing organisms to anticipate environmental conditions produced by the Earth's rotation on its axis. Here, we present the results of the manual curation of 27 genes that are associated with circadian rhythm in the genome of *Diaphorina citri*, the Asian citrus psyllid. This insect is the vector for the bacterial pathogen *Candidatus* Liberibacter asiaticus (*C*Las), the causal agent of citrus greening disease (Huanglongbing). This disease severely affects citrus industries and has drastically decreased crop yields worldwide. Based on *cry1* and *cry2* identified in the psyllid genome, *D. citri* likely possesses a circadian model similar to the lepidopteran butterfly, *Danaus plexippus*. Manual annotation will improve the quality of circadian rhythm gene models, allowing the future development of molecular therapeutics, such as RNA interference or antisense technologies, to target these genes to disrupt the psyllid biology.

**Subjects** Genetics and Genomics, Animal Genetics, Bioinformatics

## DATA DESCRIPTION

### Background

Huanglongbing (HLB), or citrus greening disease, is caused by the bacterium *Candidatus* Liberibacter asiaticus (*C*Las), which infects citrus phloem. This causes loss of fruit production and, eventually, tree death [1, 2]. The Asian citrus psyllid, *Diaphorina citri* (Order: Hemiptera; NCBI:txid121845) acts as an effective vector for the pathogen, thereby spreading the disease. Currently, there is no effective treatment to stop the disease, and it has caused widespread crop damage, resulting in substantial financial losses in the citrus industry [3]. To prevent further damages, the development of management strategies, such as molecular therapeutics, are being investigated, which requires a solid foundation of the genetic basis of psyllid biology. To better understand *D. citri* biology, we first need to identify gene pathways within the psyllid genome. To this end, genes involved in the main circadian rhythm pathway loops, along with ancillary circadian rhythm genes, were manually annotated in the *D. citri* genome. Circadian rhythm genes serve to regulate various time-based metabolic processes [4, 5]. Thus, disruption of these genes can affect various downstream biological processes. This makes many of the identified genes promising targets for the development of molecular therapeutics based on RNA interference (RNAi) strategies in insects (Table 1). Disruption of *D. citri* circadian rhythm may alter psyllid behavior, potentially hindering the spread of HLB.

### Context

The circadian rhythm is critical for an organism to regulate its biological systems in conjunction with the external environment. It is a process regulated by the interactions between multiple genes, allowing a cell to respond to, and anticipate, day/night conditions during a roughly 24-hour cycle [4, 5]. Periodicity occurs owing to fluctuations in gene expression autoregulated by inhibitory feedback loops based on external stimuli (principally light, but also temperature, humidity, nutrition, and others), also known as *Zeitgeibers* [5, 12–15]. The basic mechanism for circadian clocks in animals is highly conserved across all taxa, and the best-studied insect circadian system is that of the fruit fly, *Drosophila melanogaster* (Order: Diptera) [16]. In this model, the circadian rhythm is primarily regulated by six transcription factors: PERIOD (PER), TIMELESS (TIM), CYCLE (CYC), CLOCK (CLK), VRILLE (VRI), and PAR DOMAIN PROTEIN 1 (PDP1) [17]. The *D. melanogaster* model is an invaluable reference point for understanding the circadian rhythm, but it cannot be completely generalized to *D. citri*. This is because hemipterans are a more evolutionarily ancient lineage [18] and there are differences in cryptochrome genes (*cry*) between insects [19–21].

In nature, individual cryptochrome proteins (CRY) have evolved to complete different functions. CRY1 operates as a blue light receptor, while CRY2 works as a circadian transcription repressor [19, 21]. Insects can possess one or both cryptochrome genes, and this affects the operation of their feedback loops. These differences result in three different circadian rhythm models: the *D. melanogaster* model possesses *cry1*, the butterfly *Danaus plexippus* possesses both *cry1* and *cry2*, and the honeybee *Apis mellifera* and the beetle *Tribolium castaneum* have only *cry2* (Figure 1) [20, 21]. Cryptochrome variation gives possible insight into the evolution and function of the circadian rhythm in insects, but also makes the *D. melanogaster* model different from non-dipterans because they possess *cry2* [20].

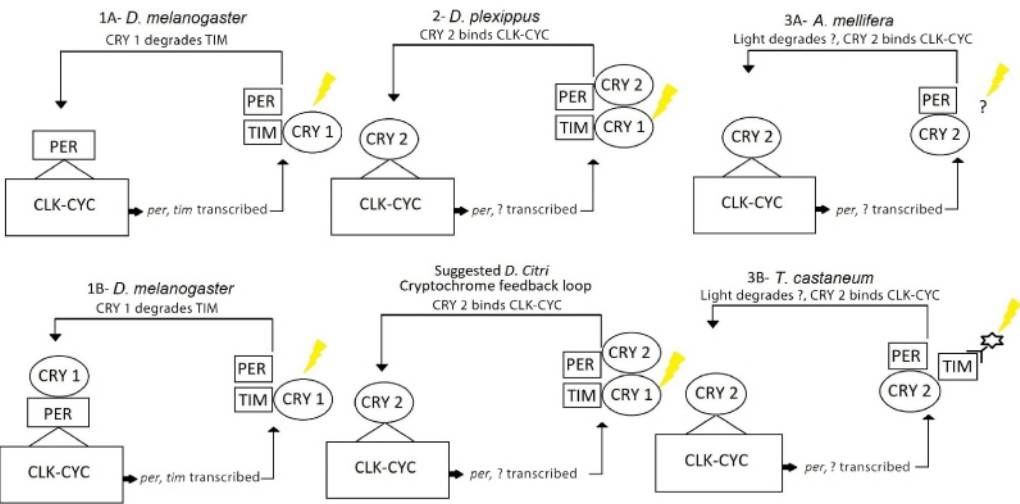

**Figure 1.** Insect clockwork pathway structure differs based on which of the two functionally different cryptochromes (CRY) are present [20]. *Drosophila melanogaster* possesses only CRY1, and there are two proposed models for its circadian pathway. (1A) CRY1 degrades the transcript of *timeless* (*tim*), and only PERIOD (PER) binds to the CLOCK-CYCLE (CLK-CYC) complex [19]. (1B) The second *D. melanogaster* model proposes that CRY1 degrades TIMELESS (TIM) then binds to PER [22]. (2) *Danaus plexippus* possesses both CRY1 and CRY2, which perform different functions [20]. The presence of both cryptochromes in *Diaphorina citri* suggests a similar circadian model to *D. plexippus*. (3) *Apis mellifera* and *Tribolium castaneum* possess only CRY2. Since CRY1 is not present to function as a blue light receptor, light interacts with alternate proteins to regulate circadian rhythm; however, CRY 2 retains the same function as in the other clockwork models. (3A) The mechanism of light interaction is unknown in *A. mellifera*. (3B) In *T. castaneum*, light enters the pathway by interacting with TIM and other associated factors (represented by the "star") [23]. The interaction of light with the pathway is represented by the lightning bolt symbol.

**Table 1.** List of annotated genes related to circadian rhythm, with the outcomes from corresponding RNA interference (RNAi) studies.

| Genes | Organism | RNAi outcome |
|---|---|---|
| *Cycle* and *Mip* | *Plautia stali* | RNAi of *cyc* in conjunction with *myoinhibitory peptide (Mip)* induced diapause by disrupting photo periodic response in ovarian development [6] |
| *Cycle* | *Riptortus pedestris* | Diapause was induced under a diapause-averting photoperiod [7] |
| *Period* | *Riptortus pedestris* | Diapause was prevented under a diapause-inducing photoperiod [7] |
| *Period* | *Ostrinia nubilalis* | Underlies a polymorphism among populations that results in the early termination of diapause [8] |
| *Pigment-dispersing factor receptor* | *Ostrinia nubilalis* | Affects diapause occurrence and may influence seasonal timing through alteration of circadian clock function [8] |
| *Clock* | *Schistocerca gregaria* | Knockdown of *clk* caused mortality in female insects before breeding age [9] |
| *Period* | *Schistocerca gregaria* | *per* and *tim* RNAi resulted in reduced progeny suggesting that PER and TIM are needed in the ovaries for egg development [9] |
| *Timeless* | *Schistocerca gregaria* | *tim* knockdown affected female insect egg deposition and resulted in reduced numbers of offspring [9] |
| *Vrille* | *Drosophila melanogaster* | Dominant maternal enhancer of defects in dorsoventral patterning causing embryo mortality [10] |
| *Takeout* | *D. melanogaster* | Mutant *takeout* causes abnormal locomotor activity resulting in insect death from starvation [11] |

In addition to cryptochrome variation between insects, the presence or absence of other genes can affect the operation of an organism's circadian model. For example, the *D. citri* genome possesses *clockwork orange* (*cwo*), *Hormone receptor 3* (*Hr3*) and *ecdysone-induced*



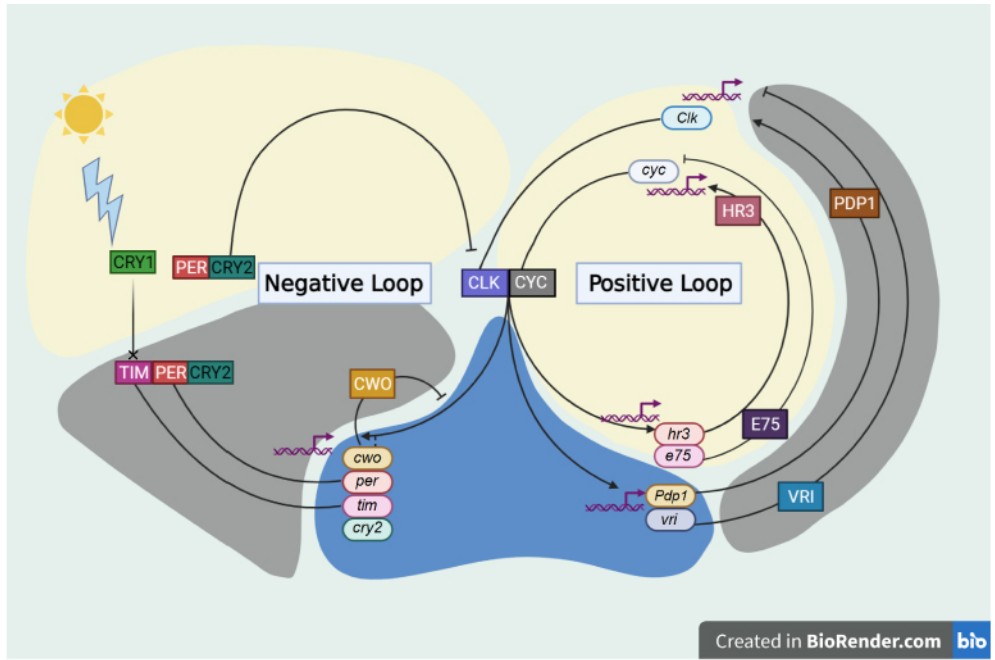

**Figure 2.** Theoretical model of circadian rhythm in *Diaphorina citri* based on curated genes and studies in other insect species. Ovals represent genes; boxes represent proteins (with the exception of "Positive Loop" and "Negative Loop" which are labels); connected boxes indicate that a protein complex has formed; lines from boxes to ovals ending in arrows represent that a protein is inducing transcriptional activation of a gene; lines from boxes to ovals ending in perpendicular lines represent that a protein is repressing transcriptional activation of a gene; lines connecting ovals to boxes indicate that a gene was translated into a protein; lines ending in an "X" indicate that degradation is occurring; helical symbols indicate that transcription occurs; the sun and lightning bolt symbol represent the interaction of light with the pathway; the colors of the background correspond to the time of day that a part of the process occurs (yellow: day, gray: night, blue: dusk). List of abbreviations: *Clk: clock;* CRY1: Cryptochrome 1; *cry2: cryptochrome 2; cwo; clockwork orange; cyc: cycle; e75: ecdysone-induced protein 75; hr3: hormone receptor 3; Pdp1: par domain protein 1; per: period; tim: timeless; vri: vrille.* Figure created using BioRender.com [33].

*protein 75* (*E75*), which suggests a *cyc* transcriptional regulation system similar to those found in *Thermobia domestica* and *Gryllus bimaculatus* [24, 25].

Based on the genes identified and annotated in the *D. citri* genome, Figure 2 presents a theoretical model of the circadian rhythm pathway in *D. citri*. Regulation of the pathway operates in two feedback loops; one negative and one positive [17]. The negative loop involves the *per* and *tim* genes, whose expression is promoted when a heterodimer of CLK and CYC binds to their E-Box promoters [16, 26–29]. The CLK–CYC heterodimer is also responsible for the transcriptional activation of *cry2* and *cwo* [25]. Transcripts for *tim* and *per* peak at dusk, allowing their protein products to accumulate in the cytoplasm during the night [16, 25, 30]. A trimer of PER, TIM and CRY2 forms and migrates to the nucleus. CRY1 acts as a blue light photoreceptor, which in turn degrades TIM, resulting in a PER–CRY2 dimer [31]. PER primarily serves to stabilize CRY2, which acts as an important transcriptional repressor, preventing CLK–CYC from transcribing *per* and *tim* genes, resulting in mRNA levels being minimum at dawn [31, 32]. Additionally, during the night, CWO acts to repress *per* and *tim* transcription through competitive inhibition, binding to *per* and *tim* E-box promoters, preventing CLK–CYC from inducing expression [25].

In the positive loop, *Clk* and *cyc* are rhythmically expressed. Transcription of *vri* and *Pdp1* occur through activation by CLK—CYC [16]. VRI acts as a transcriptional repressor and

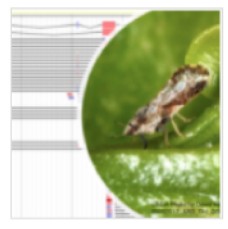

Annotating genes in Diaphorina citri genome version 3

Teresa Shippy[1], S Miller[2], C Massimino[3], C Vosburg [Indian River State College[4]. PS Hosmani[5]. M Flores-Gonzalez[5]. LA...
[1]Kansas State University, [2]Kansas State University, Allen County Community College. [3]Indian River State College....

Dec 16, 2020

**Figure 3.** Protocol for *Diaphorina citri* genome community curation [37]. https://www.protocols.io/widgets/doi?uri=dx.doi.org/10.17504/protocols.io.bniimcce

acts to delay the action of PDP1, which acts as a transcriptional activator [16, 17]. Expression of *Clk* by PDP1 occurs at dawn, causing *Clk* transcripts to peak in concentration in the opposite phase to that of *per* and *tim* [16]. Rhythmic regulation of *cyc* transcription was observed to occur in a similar manner, with E75 acting as a transcriptional repressor which that delays expression of *cyc* by HR3 [24, 25].

## METHODS

Through a community genome curation project, the *D. citri* (NCBI:txid121845) genes involved in the circadian rhythm pathway were manually annotated in genome version 3.0 [34–36], following our standard annotation protocol (Figure 3) [37]. Protein sequences of circadian rhythm genes were collected from the National Center of Biotechnology (NCBI) protein database (RRID:SCR_003496) [38]. The collected protein sequences were used to BLAST (Basic Local Alignment Search Tool) search the *D. citri* MCOT (Maker (RRID:SCR_005309), Cufflinks (RRID:SCR_014597), Oases (RRID:SCR_011896), and Trinity (RRID:SCR_013048)) protein database for MCOT model identifiers [39]. The circadian rhythm genes were found in the *D. citri* genome (version 3.0) using the mapped MCOT model identifiers in the WebApollo (RRID:SCR_005321) system hosted at the Boyce Thompson Institute. Utilizing the NCBI BLAST tool (RRID:SCR_004870), a reciprocal BLASTp was performed for each located *D. citri* gene model to confirm the predicted circadian rhythm gene. Examination of RNA-seq reads, Illumina DNA-seq reads, and PacBio Iso-seq transcripts were used to annotate the final *D. citri* circadian rhythm genes. These tools allowed common issues within gene models to be identified and corrected, such as artifact duplications, splice site errors, incorrect untranslated regions, insertions, deletions, incorrect start sites, incorrect stop sites, faulty exon boundaries, and alternative splice sites. These issues arise from sequencing or genome assembly errors and analysis of available evidence makes necessary revisions possible. A more detailed explanation of the annotation process is available through a previously reported protocol (Figure 3) [37]. After correcting any of the common issues previously described, the manually annotated circadian rhythm gene models were included into the version 3.0 Official Gene Set (OGS).

Table 2 shows the evidence supporting the *D. citri* final gene models [39]. Comparison of data from the MCOT transcriptome, de novo transcriptome, PacBio Iso-seq transcripts, and RNA-seq reads and the annotated *D. citri* circadian rhythm gene determined the status of support listed in Table 2. A gene copy number table was created to observe the number of true duplications of circadian rhythm genes of interest that were present within other insect species (Table 3). Creation of this table involved conducting peptide BLASTs for each

**Table 2.** Evidence supporting gene annotation available as tracks in the Apollo genome curation tool. There are 27 annotated circadian rhythm genes in *Diaphorina citri*. Each gene model has been assigned an Official Gene Set (OGS) version 3.0 gene identifier. Evidence types used to validate or modify the structure of the gene model have been marked with an X if available and are left blank if unavailable. Genes marked as complete have been marked with an X and have a complete coding region. Those not marked as complete are left blank and were only annotated as partial gene models. Completeness refers to the wholeness of the coding regions of these models and was determined through comparison with orthologous proteins. Descriptions of the various evidence sources and their strengths and weaknesses are included in the online protocol [37]. All evidence supporting annotation is available from the Citrus Greening Solutions website [39]. Terminology is used as in Flybase [41].

| Gene | OGSv3 ID | MCOT ID | *de novo* | Gene model Complete/Partial | MCOT | Iso-seq | RNA-seq | Ortholog |
|---|---|---|---|---|---|---|---|---|
| **D. melanogaster model core clock genes** | | | | | | | | |
| *Circadian locomotor output cycles protein kaput* | Dcitr05g05480.2.1 | MCOT03010.0.CO | X | | X | | X | X |
| *Cycle* | Dcitr05g16440.2.3 | | | X | | | X | X |
| *PAR domain protein 1* | Dcitr05g02780.2.1 | MCOT14609.2.CT | | X | X | | X | X |
| *Period* | Dcitr09g03540.1.1 | MCOT12213.0.CO | | X | X | | X | |
| *Timeless* | Dcitr04g02940.1.1 | MCOT23350.0.CT | X | X | X | X | X | X |
| *Vrille* | Dcitr10g05230.1.1 | MCOT16283.2.CT | X | X | X | X | | |
| **Other negative loop-associated genes** | | | | | | | | |
| *Casein kinase 2 alpha subunit* | Dcitr02g18000.1.1 | MCOT21500.0.CO | X | X | X | | X | X |
| *Casein kinase 2 beta subunit* | Dcitr11g02180.1.1 | | | X | X | X | | X |
| *Cryptochrome 1* | Dcitr07g01550.1.1 | MCOT00625.0.CT | X | X | X | X | X | X |
| *Cryptochrome 2* | Dcitr09g09100.1.1 | MCOT19207.3.CO | X | X | X | | X | X |
| *Double-time* | Dcitr07g07760.2.1 | MCOT08150.0.MT | | X | X | | X | X |
| **Other positive loop-associated genes** | | | | | | | | |
| *Ecdysone-induced protein 75* | Dcitr12g07770.2.1 | MCOT02379.0.CT | | X | X | X | X | X |
| *Hormone Receptor 3/Rora* | Dcitr08g04060.1.1 | MCOT23147.0.CC | | X | X | X | | X |
| **Genes documented to be under circadian influence** | | | | | | | | |
| *Clockwork orange isoform 1* | Dcitr01g06780.1.1 | MCOT17066.1.CO | | X | X | | X | X |
| *Clockwork orange isoform 2* | Dcitr01g06780.1.1 | MCOT17066.2.CO | X | X | X | | X | X |
| *Ecdysone receptor* | Dcitr02g19530.1.1 | MCOT15763.0.CT | | X | X | X | X | X |
| *Hormone Receptor 3 Homolog* | Dcitr08g02450.1.3 | | X | X | | | | X |
| *Insulin related peptide* | Dcitr06g08750.1.1 | MCOT08282.0.MO | | X | X | | X | |
| *Lark* | Dcitr01g16940.1.1 | MCOT03363.0.CT | X | X | X | X | X | X |
| *Ovary maturing parsin* | Dcitr09g03370.1.1 | MCOT04488.1.CT | | X | X | | | |
| *Protein phosphatase PP2A 55 kDa regulatory subunit* | Dcitr09g05720.1.1 | MCOT08076.0.OO | X | X | X | X | X | X |
| *Serine/Threonine-Protein phosphatase 2A 56 kDa regulatory subunit epsilon* | Dcitr03g10950.1.1 | MCOT02984.0.CT | X | X | X | X | X | X |
| *Serine/Threonine-Protein phosphatase 2A 56 kDa regulatory subunit epsilon* | Dcitr03g03510.1.1 | MCOT08848.0.MO | X | X | X | | X | X |
| *Takeout 1* | Dcitr07g05480.1.3 | MCOT21001.2.CT | X | X | X | X | X | X |
| *Takeout 2* | Dcitr06g07220.1.1 | MCOT16413.0.CO | | X | X | X | X | X |
| *Takeout-Like* | Dcitr09g07620.1.1 | MCOT13357.0.CC | X | X | X | X | X | X |
| *Timeless 2/timeout fragment 1* | Dcitr00g11340.2.1 | | | | | | X | X |
| *Timeless 2/timeout fragment 2* | Dcitr00g05520.2.1 | | | | | | | X |
| *Timeless 2/ timeout Fragment 3* | Dcitr00g05510.2.1 | | | | | | X | X |
| *Vestigial* | Dcitr08g07920.1.1 | MCOT06810.0.CC | X | X | X | | X | X |
| *Vitellogenin-1* | Dcitr08g09320.1.1 | | | X | | X | | |
| *Vitellogenin-2* | Dcitr01g05660.1.1 | MCOT05282.2.CO | | X | X | X | X | |

OGS v3.0 *D. citri* model. The NCBI descriptions, percentage sequence identity, and query coverage for each gene were observed. Only orthologs with appropriate descriptions with a 40% or higher identity and query coverage were incorporated as evidence. Table 4 displays the ortholog accession numbers used to create Table 3. A neighbor-joining phylogenetic tree (Figure 4) of the *D. citri* CRY1 and CRY2 protein sequences was made, along with various related orthologs, with MEGA 7 (MEGA software, RRID:SCR_000667) [40]. Tree construction used MUSCLE (RRID:SCR_011812) for multiple sequence alignments with p-distance for

**Table 3.** Gene copy numbers for *Diaphorina citri* compared with other insects. Numbers were obtained from NCBI unless otherwise indicated, except for *D. melanogaster*, whose copy numbers were obtained from Flybase [41].

| Gene name | Diaphorina citri | Acyrthosiphon pisum | Bemisia tabaci | Cimex lectularius | Halyomorpha halys | Drosophila melanogaster | Anopheles aegypti |
|---|---|---|---|---|---|---|---|
| **D. melanogaster model core clock genes** | | | | | | | |
| Circadian locomotor output cycles protein kaput | 1 | 1* | 0 | 1 | 0 | 1 | 0 |
| Cycle | 1 | 1* | 1 | 1 | 1 | 1 | 1 |
| PAR domain protein 1 | 1 | 1* | 1 | 1 | 0 | 1 | 0 |
| Period | 1 | 1* | 1 | 1 | 1 | 1 | 1 |
| Timeless | 1 | 1* | 1 | 1 | 1 | 1 | 1 |
| Vrille | 1 | 1* | 0 | 0 | 0 | 1 | 0 |
| **Other negative loop-associated genes** | | | | | | | |
| Casein kinase 2 alpha subunit | 1 | 1* | 1 | 0 | 2 | 1 | 1 |
| Casein kinase 2 beta subunit | 1 | 1* | 1 | 1 | 1 | 1 | 1 |
| Cryptochrome 1 | 1 | 1* | 1 | 0 | 0 | 1 | 3 |
| Cryptochrome 2 | 1 | 2* | 1 | 0 | 0 | 0 | 1 |
| double-time | 1 | 1* | 1 | 1 | 0 | 0 | 0 |
| **Other positive loop-associated genes** | | | | | | | |
| Ecdysone-induced protein 75 | 1 | 1 | 1 | 1 | 1 | 1 | 1 |
| Hormone Receptor 3 | 1 | 0 | 1 | 1 | 2 | 1 | 1 |
| **Genes documented to be under circadian influence** | | | | | | | |
| Clockwork orange | 1 | 0 | 0 | 0 | 1 | 1 | 1 |
| Ecdysone receptor | 1 | 0 | 1 | 1 | 1 | 1 | 1 |
| Hormone Receptor 3 Homolog | 1 | 0 | 0 | 0 | 0 | 0 | 0 |
| Insulin related peptide | 1 | 0 | 1 | | 1 | 0 | 1 |
| Lark | 1 | 1 | 1 | 0 | 1 | 1 | 1 |
| Ovary maturing parsin | 1 | 0 | 0 | 0 | 0 | 0 | 0 |
| Protein phosphatase PP2A 55 Kda regulatory subunit | 1 | 1 | 1 | 1 | 2 | 0 | 1 |
| Serine/Threonine Protein phosphatase 2A 56 Kda regulatory subunit epsilon | 2 | 1 | 1 | 1 | 1 | 0 | 1 |
| Takeout | 2 | 3 | 0 | 1 | 11 | 1 | 8 |
| Takeout-Like | 1 | 7 | 12 | 19 | 23 | 0 | 1 |
| Timeout | 1 | 0 | 1 | 1 | 1 | 1 | 1 |
| Vestigial | 1 | 0 | 1 | 1 | 1 | 1 | 1 |
| Vitellogenin-1 | 1 | 1 | 1 | 1 | 2 | 0 | 2 |
| Vitellogenin-2 | 1 | 0 | 0 | 0 | 0 | 0 | 0 |

*Denotes gene copy numbers retrieved from Cortes *et al.* 2010 [16].

determining branch length and 1000 bootstrap replicates [40]. The insects, with their corresponding accession numbers, used to create the phylogenetic tree are found in Table 5. The circadian rhythm pathway image (Figure 2) was created using BioRender (RRID:SCR_018361) [33].

## DATA VALIDATION AND QUALITY CONTROL
### Core clock genes
Gene models were annotated in the Apollo *D. citri* v3.0 genome using the Citrus Greening online portal [42, 43]. We identified all six of the core clock genes found in the *D. melanogaster* model [11]: *Clk, cyc, per, tim, vri* and *Pdp1*. As has been reported in *D. melanogaster* and the hemipteran *A. pisum*, only one copy of each gene was found in the *D. citri* genome (Table 3). However, false duplications of *Clock* and *Pdp1* resulting from genome assembly errors were found and removed from the OGS. These errors are identifiable within WebApollo by examining the Illumina DNA-seq or RNA-seq data for the

**Table 4.** Accession numbers searched using NCBI for ortholog copy number analysis (see Table 3). Genes without listed accession numbers underneath the corresponding Hemiptera did not meet the criteria listed in the Methods section, or had no identifiable NCBI ortholog accession number.

| Genes | *Acyrthosiphon pisum* | *Nilaparvata lugens* | *Sipha flava* | *Bemisia tabaci* | *Halomorpha halys* | *Cimex lectularius* |
|---|---|---|---|---|---|---|
| *Circadian locomotor output cycles protein kaput* | NP_001164531.1 | NP_001164531.1 | NP_001164531.1 | NP_001164531.1 | NP_001164531.1 | NP_001164531.1 |
| *Cycle* | XP_008187055.1 | XP_039286553.1 | XP_025408235.1 | XP_018917520.1 | XP_024218924.1 | XP_024083945.1 |
| *PAR domain protein 1* | ARM65425.1 | XP_039287503.1 | XP_025423858.1 | XP_018914309.1 | XP_014272589.1 | XP_024080389.1 |
| *Period* | | | | | | |
| *Timeless* | | XP_039278663.1 | XP_025421933.1 | XP_018904703.1 | XP_014278196.1 | XP_014260027.1 |
| *Vrille* | | | | | | |
| *Casein kinase 2 alpha subunit* | XP_001942962.2 | XP_039276357.1 | XP_025423328.1 | XP_018898827.1 | XP_014287349.1 | XP_014255415.1 |
| *Casein kinase 2 beta subunit* | NP_001119697.1 | XP_039288290.1 | XP_025409432.1 | XP_018916320.1 | XP_014270634.1 | XP_014251170.1 |
| *Cryptochrome 1* | NP_001164532.2 | XP_039290665.1 | XP_025414102.1 | XP_018915448.1 | XP_014279454.1 | XP_014255347.1 |
| *Cryptochrome 2* | NP_001164573.1 | AJY53622.1 | XP_025423877.1 | XP_018916709.1 | XP_014279454.1 | XP_014255347.1 |
| *Double-time* | XP_008186371.1 | XP_039290506.1 | XP_025413925.1 | XP_018907315.1 | XP_024216975.1 | XP_014260368.1 |
| *Ecdysone-induced protein 75* | XP_016657682.1 | XP_039292205.1 | XP_025405680.1 | XP_018907602.1 | XP_014276625.1 | XP_014240380.1 |
| *Hormone Receptor 3 /Rora* | XP_008189753.1 | XP_039297195.1 | XP_025413275.1 | QJD20722.1 | XP_024216207.1 | XP_014245664.1 |
| *Clockwork orange isoform 1* | XP_003245398.1 | XP_022192550.2 | XP_025417490.1 | XP_018913029.1 | | |
| *Clockwork orange isoform 2* | XP_003245398.1 | XP_022192550.2 | XP_025417492.1 | XP_018913029.1 | XP_014281247.1 | XP_024080847.1 |
| *Ecdysone receptor* | XP_008182215.1 | XP_039296403.1 | XP_025415323.1 | XP_018906416.1 | XP_014281746.1 | XP_024081622.1 |
| *Hormone Receptor 3 Homolog* | XP_008189753.1 | XP_039297191.1 | XP_025413279.1 | XP_018918044.1 | XP_024216205.1 | XP_014245659.1 |
| *Insulin related peptide* | | | | | | |
| *Lark* | XP_008183373.1 | XP_022195298.2 | XP_025417052.1 | XP_018904452.1 | XP_014279091.1 | XP_014259770.1 |
| *Ovary maturing parsin* | | | | | | |
| *Protein phosphatase PP2A 55 kDa regulatory subunit* | XP_003243934.1 | XP_022199102.1 | XP_025406298.1 | XP_018906142.1 | XP_024214848.1 | XP_014252979.1 |
| *Serine/Threonine-Protein phosphatase 2A 56 kDa regulatory subunit epsilon* | XP_001944239.2 | XP_039283040.1 | XP_025414701.1 | XP_018914200.1 | XP_024218201.1 | XP_014249434.1 |
| *Serine/Threonine-Protein phosphatase 2A 56 kDa regulatory subunit epsilon* | XP_016656660.1 | XP_039282421.1 | XP_025409028.1 | XP_018915172.1 | XP_014285453.1 | XP_014249435.1 |
| *Takeout 1* | XP_001950706.1 | XP_022197942.1 | XP_025408661.1 | XP_018909796.1 | XP_024218715.1 | XP_014241740.1 |
| *Takeout 2* | XP_001950829.1 | | XP_025408664.1 | XP_025408664.1 | XP_024218715.1 | XP_001950829.1 |
| *Takeout-Like* | XP_001947537.1 | XP_022184393.2 | XP_025422121.1 | XP_018917268.1 | XP_014272244.1 | XP_014254912.1 |
| *Timeless 2/timeout fragment 1* | XP_008186746.1 | XP_039290694.1 | XP_025415567.1 | XP_018907854.1 | XP_024217293.1 | XP_014246467.1 |
| *Timeless 2/timeout fragment 2* | XP_008186746.1 | XP_039290694.1 | XP_025415562.1 | XP_018907854.1 | XP_024217293.1 | XP_014246467.1 |
| *Timeless 2/timeout Fragment 3* | XP_008186746.1 | XP_039290694.1 | XP_025415564.1 | XP_018907854.1 | XP_024217293.1 | XP_014246467.1 |
| *Vestigial* | XP_003242605.1 | AKE98361.1 | XP_025418544.1 | XP_018900015.1 | XP_014275299.1 | XP_014249102.1 |
| *Vitellogenin-1* | | | | | | |
| *Vitellogenin-2* | | | | | | |

locus and by conducting NCBI pairwise BLAST comparisons between duplicate and true models. Additionally, two of the models, *per* and *tim*, were substantially improved from the computational gene predictions. Annotation of *per* identified and corrected a splice site error, increased the amino acid length of the peptide, and improved BLAST score, query coverage and percent identity. Curation of *tim* also led to an amino acid increase from 1093 to 1109 and saw notable improvements in BLAST score, query coverage and percentage sequence identity. Notably, the added amino acids were part of a Timeless serine-rich domain in the 260–292 amino acid range of the peptide, which has been identified as an important site for phosphorylation [44]. Annotation of the *vri* model may prove particularly important since lack of VRI function leads to embryonic mortality in *D. melanogaster* (Table 1) [10]. This may make *vri* a useful target for *D. citri* population control.



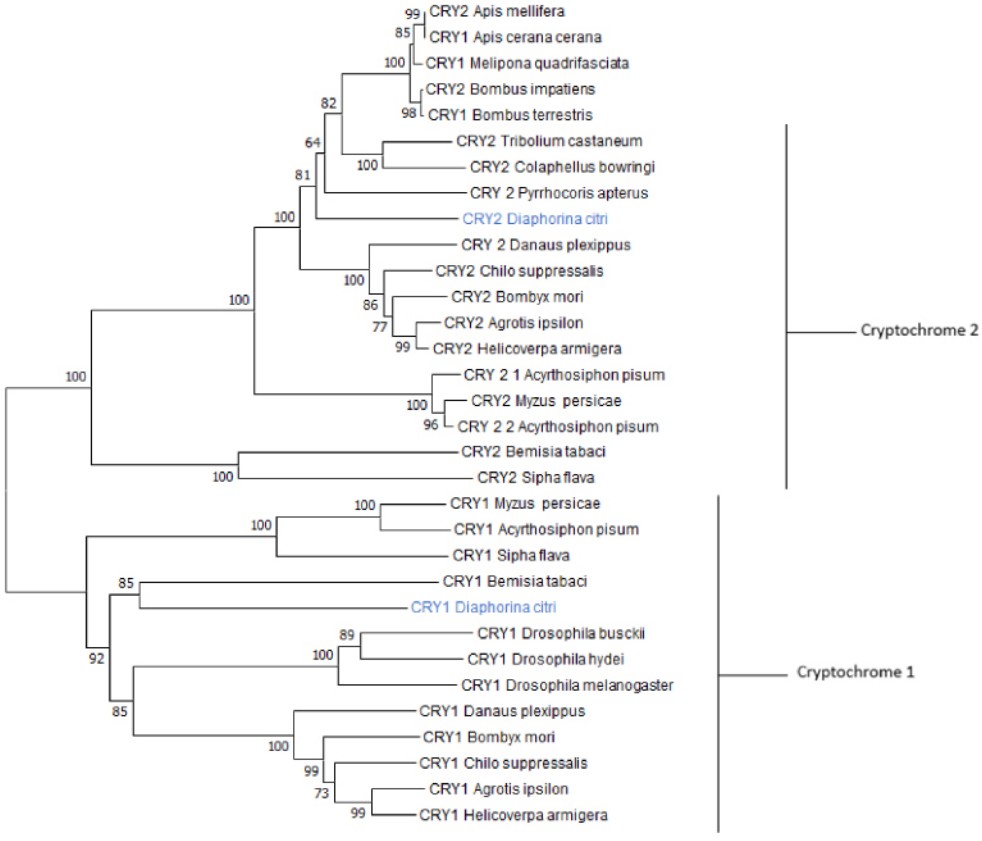

**Figure 4.** Phylogenetic tree of Cryptochrome 1 and 2 amino acid sequences. _Diaphorina citri_ cryptochromes are in blue and group with their respective type of cryptochrome. _Nephila clavipes_ CRY2 was used as the outgroup. The tree was made with MEGA7 using the bootstrap test (1000) replicates with the percentage of replicate trees clustered together shown next to the branches. The total number of positions used for alignment was 349 using MUSCLE alignment in MEGA7 [40].

## Other negative loop-associated genes

Genes related to the negative feedback loop, aside from the core clock genes, include _cry1,_ _cry2, Casein kinase 2 alpha (Ck2α) subunit, Casein kinase 2 beta (Ck2β) subunit_, and _double-time (dbt)_ (Table 3). Cryptochrome copy number variation (Figure 1) is of great importance since discrepancies between insects gives possible insight into the evolution and function of circadian rhythm in insects [20]. In _D. citri_, both _cry1_ and _cry2_ genes were found. A pairwise alignment between the two annotated models resulted in 41% amino acid identity with 93% query coverage. Both models have the DNA_photolyase (pfam00875) and FAD_binding_7 (pfam03441) domains typically found in CRY proteins. The DNA_photolyase domain is the binding site for a light-harvesting cofactor, and the FAD_binding_7 domain binds a flavin adenine dinucleotide (FAD) molecule [45]. The BLAST results confirm that the two models were not duplications, and phylogenetic analysis could discern the identity of each cryptochrome (Figure 4). The presence of _cry1_ and _cry2_ genes in _D. citri_ reinforces that non-drosophilid insects possess _cry2_ [20]. The presence of both genes also suggests that _D. citri_ follows an ancestral butterfly-like clockwork model (Figure 1) [20]. _D. citri_ only has one _cry2_ gene, while _A. pisum_ has two _cry2_ genes; this supports the assertion that _A. pisum_ core clock genes may be evolving at a faster rate than in other hemipterans [16].

**Table 5.** NCBI accession numbers used to create the neighbor-joining phylogenetic tree, displaying orthologous sequences for CRY1 and CRY2 genes.

| Insects | CRY1 | CRY2 |
|---|---|---|
| **Non-hemipteran** | | |
| *Drosophila melanogaster* | NP_732407.1 | |
| *Danaus plexippus* | XP_032522602.1 | XP_032511499.1 |
| *Drosophila busckii* | XP_017845052.1 | |
| *Drosophila hydei* | XP_023171160.2 | |
| *Apis mellifera* | | NP_001077099.1 |
| *Apis cerana* | PBC28621.1 | |
| *Melipona quadrifasciata* | KOX67887.1 | |
| *Bombus impatiens* | | NP_001267051.1 |
| *Bombus terrestris* | XP_003398531.1 | |
| *Tribolium castaneum* | | NP_001076794.1 |
| *Colaphellus bowringi* | | APP94027.1 |
| *Agrotis ipsilon* | AFJ22638.1 | AFJ22639.1 |
| *Helicoverpa armigera* | XP_021184238.1 | ADN94465.2 |
| *Bombyx mori* | NP_001182628.1 | NP_001182627.1 |
| *Chilo suppressalis* | CDK02014.2 | AHL69753.1 |
| **Hemipteran** | | |
| *Bemisia tabaci* | XP_018915448.1 | XP_018904762.1 |
| *Sipha flava* | XP_025414102.1 | XP_025423877.1 |
| *Myzus persicae* | AUN43313.1 | AUN43314.1 |
| *Acyrthosiphon pisum* | NP_001164532.2 | NP_001164572.1 NP_001164573.1 |
| *Pyrrhocoris apterus* | | AGI17567.1 |

Another important gene in the negative loop is *dbt*, also known as *discs overgrown* (*dco*). This gene encodes a circadian rhythm regulatory protein that affects the stability of PER [46] and was also annotated. The mammalian homolog of *dbt* is *Casein kinase 1* (*Ck1*) *epsilon* [47]. The *Homo sapiens* (ABM64212.1) and *D. melanogaster* (NP_733414.1) homologs both have the STKc_CK1_delta_epsilon (accession number: cd14125) domain at the same position, amino acids 8–282. The *D. citri* model reflects the conservation between species by also exhibiting a CK1_delta_epsilon domain located at the same position.

## Other positive loop-associated genes

Two genes present in *D. citri*, *Hr3* and *E75*, are responsible for positive loop transcriptional regulation of *cyc* in *T. domestica* and *G. bimaculatus* [24, 25], and a similar regulation mechanism may exist in *D. citri* (Figure 2). Curation of these genes led to minimal changes to the *Hr3* gene model; however, changes in *E75* resulted in its translated peptide sequence being extended from 664 to 729. This improved its score, percentage sequence identity, and query coverage when BLASTed to orthologous insects.

## Genes documented to be under circadian influence

Fourteen additional genes involved in the circadian rhythm, but not directly classified in the two feedback loops, were also annotated (Table 3). Of these genes, *takeout* is of interest as a potentially good molecular target to control *D. citri*. *Takeout* in *D. melanogaster* has been shown to regulate starvation response, and if the gene or protein loses function, the organism dies faster in starvation conditions than the wild type (Table 1) [11]. *Takeout* and *takeout-like* in *D. citri* have low gene copy numbers compared with other non-drosophilid insects (Table 3). The discrepancy may be because these genes are computationally



predicted in the other insects, and many sequences are short and might represent fragments of larger genes.

Another of these ancillary genes annotated was the *tim* paralog *timeout*, also known as *timeless 2* [23]. *Timeout* has several functions and does not fit categorically into either of the feedback loops, although it is structurally very similar to *tim* [48]. To confirm that the annotated gene models were not the result of a false duplication in the genome assembly, they were compared with orthologs from other insects. The conserved domains in protein sequences for *tim* and *timeout* from *D. melanogaster* (NP_722914.3 and NP_524341.3) and *A. pisum* (ARM65416.1 and XP_016662949.1) were compared with the *D. citri* sequences. For the TIM proteins, the main conserved domain was the TIMELESS (pfam04821) domain located near the N-terminus. TIMEOUT proteins also have this N-terminal TIMELESS domain, but also have a C-terminal TIMELESS_C (pfam05029) domain. These conserved domains were also identified in the corresponding *D. citri* protein sequences, indicating that the *timeout* model is not an artifactual duplication of *tim*. In genome version 3.0, a problem still exists causing the *timeout* sequence to be split into three different models (Table 2), despite the comparative sequence analysis that supports the presence of this gene in the genome.

## CONCLUSION

The circadian rhythm is an important pathway for an organism's ability to regulate its biological systems in conjunction with the external environment [5]. A total of 27 putative circadian rhythm-associated genes have been annotated in the *D. citri* genome (Table 2). These data have been used to construct models of both positive and negative feedback loops. The relationship of these genes to *D. citri* circadian rhythm is putative and is based on insect orthologs. Future research with RNA-seq experimental analysis over a 24-hour day/night cycle will validate these relationships and provide a more definitive list of circadian rhythm associated genes in *D. citri*. Annotation of these genes provides a better understanding of the circadian pathway in *D. citri* and adds to the evidence supporting the ways in which hemipteran circadian rhythm pathways function.

## REUSE POTENTIAL

Manual curation of these circadian rhythm genes was conducted as part of the *D. citri* collaborative community annotation project [43, 49]. These models will be incorporated into the third OGS of the Citrus Greening Expression Network (CGEN) [43]. This publicly available tool is useful for comparative expression profiling through its transcriptome data for different *D. citri* tissues and life stages. There is considerable potential for the practical application of these curated models in controlling the spread of Huanglongbing. Understanding the circadian pathway and essential genes within it provide opportunities for pest control strategies through molecular-based therapeutics. Availability of accurate gene models brought about through our manual annotations can facilitate the design of future experiments aimed at developing gene editing and RNAi systems to target and disrupt critical *D. citri* biological processes.

## DATA AVAILABILITY

The *D. citri* genome assembly, official gene sets, and transcriptome data are accessible on the Citrus Greening website [39]. The gene models will also be part of an updated OGS

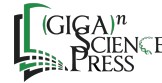

version 3 for *D. citri* [35, 37, 42]; the data are also available through NCBI (BioProject: PRJNA29447). All additional data supporting this article is available via the *GigaScience* GigaDB repository [50].

## EDITOR'S NOTE

This article is one of a series of Data Releases crediting the outputs of a student-focused and community-driven manual annotation project curating gene models and, if required, correcting assembly anomalies, for the *Diaphorina citri* genome project [51].

## DECLARATIONS
## LIST OF ABBREVIATIONS

BLAST: Basic local alignment search tool; BLASTp: BLAST protein*;* CGEN: Citrus Greening Expression Network; *Ck1*: *Casein kinase 1*; *clk*: *Clock*; *CLOCK*: *Circadian locomotor output cycles protein kaput*; *C*Las: *Candidatus* Liberibacter asiaticus; *cry1*: *Cryptochrome 1*; *cry2*: *Cryptochrome 2*; *cwo*; *clockwork orange*; *cyc*: *cycle*; *dbt*: *double-time*; *dco*: *discs overgrown*; *E75*: *ecdysone-induced protein 75*; HLB: Huanglongbing; *Hr3*: *Hormone receptor 3*; Iso-seq: Full-length isoform sequencing; MCOT: MAKER Cufflinks Oases Trinity; *Mip: myoinhibitory peptide*; NCBI: National Center for Biotechnology Information; OGS: Official gene set; *Pdp1*: *PAR domain protein 1*; *per*: *period*; PP2A: *Protein phosphatase 2A*; RNA-seq: RNA-sequencing; RNAi: RNA interference; STKc: Serine/threonine kinase catalytic domain; *tim*: *timeless*; *vri*: *vrille*.

## ETHICAL APPROVAL

Not applicable.

## CONSENT FOR PUBLICATION

Not applicable.

## COMPETING INTERESTS

The authors declare that they have no competing interests.

## FUNDING

This work was supported by USDA-NIFA grants 2015-70016-23028, HSI 2020-38422-32252 and 2020-70029-33199.

## AUTHORS' CONTRIBUTIONS

SJB, WBH, TD and LAM conceptualized the study; TD, SS, TDS, CM, JAQ, CV and SJB supervised the study; TD, SS, LAM and SJB contributed to project administration; LD, CM, JN, YO, ME, ND, RM, AN, RH, KG, MK, MTH conducted the investigation; PH, SS, MF-G contributed to software development; SS, TDS, PH and MF-G developed methodology; SJB, WBH, KP-S, JAQ, TD and LAM acquired funding; MR, LD, TP and YO prepared and wrote the original draft; LD, TP, WBH, TDS, JAQ, JBB, YO, ME, RM, AN, reviewed and edited the draft.

## ACKNOWLEDGEMENTS

We would like to thank Helen Wiersma-Koch (Indian River State College), Elizabeth Michels, Sarah Michels, and Tanner Wise for their assistance.

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
