## [Reviewer Report]

Comments on revised manuscriptThanks to the authors for their detailed answers. Most of the unclear points were well supplemented in the revised manuscript, though there are two minor details to be aware of.

Is the data acquisition clear, complete and methodologically sound? No
Additional Comments: Is there sufficient detail in the methods and data-processing steps to allow reproduction? No
Additional Comments: The method section of this paper seems too brief and should be supplemented in detail according to these studied rhythm genes.
We apologise for this oversight as we believe strongly in open access and reproducibility. We have completely rewritten the method and have added Table 4 and 5. Table 4 was lists the NCBI accession numbers of Hemiptera used for ortholog support described in Table 2. Table 5 lists all insects (Non-hemipteran and Hemipteran) and their NCBI accession numbers used in creation of the phylogenetic tree and referred to in methods section.

1. Table 4 and 5 take a large place in main manuscript and perhaps better as supplementary tables.

Table 2：“X” and “space” should be explained in the table notes. In addition, the analysis methods of “MCOT, ISO-SEQ,RNA-Seq and Ortholog” in the "Evidence Supporting Annotation" are not described in this article. Are these results collected from an existing online database or generated from new analysis that author done in this study?
We now provide a citation for citrusgreening.org as the source of the evidence used for supporting annotation. We also explain the use of “X” and “blank” in the figure legend as requested. After collection of ortholog accessions from NCBI for Table 4, we reviewed Table 2 and found some genes were missing the “X” indicating ortholog support. We have added ortholog support for the following genes: circadian locomotor output cycles protein kaput, PAR domain protein 1, Cryptochrome 2, double-time, Hormone Receptor 3 /Rora, clockwork orange isoform 1 and clockwork orange isoform 2. We also removed ortholog support for two genes (period, Vitellogenin-1) as they did not meet the BLASTp result criterion explained in the methods section.

2. I'm a little confused about the ortholog criterion. How do you explain the criterion of “40.00 identity”? Is it summarized from your experiment? Is each gene aligned with at least one reference sequence (in Table 4) over 40.00 alignment identity defined as ortholog gene?

---

## [Reviewer Report]

Reviewer name and names of any other individual's who aided in reviewer Mary Ann TuliDo you understand and agree to our policy of having open and named reviews, and having your review included with the published papers. (If no, please inform the editor that you cannot review this manuscript.)YesIs the language of sufficient quality?YesPlease add additional comments on language quality to clarify if needed
NAAre all data available and do they match the descriptions in the paper? YesAdditional CommentsThe author states, "The gene models will also be part of an updated OGS version 3 for D. citri". I am wondering when this updated version will be available.
In addition the author states, "the data is also available through NCBI (BioProject: PRJNA29447". It would be good to include the GenBank accession numbers of the 27 D. citri genes.Are the data and metadata consistent with relevant minimum information or reporting standards? See GigaDB checklists for examples <a href="http://gigadb.org/site/guide" target="_blank">http://gigadb.org/site/guide</a>YesAdditional CommentsIs the data acquisition clear, complete and methodologically sound?YesAdditional CommentsIs there sufficient detail in the methods and data-processing steps to allow reproduction?YesAdditional CommentsIs there sufficient data validation and statistical analyses of data quality? YesAdditional CommentsIs the validation suitable for this type of data?YesAdditional CommentsIs there sufficient information for others to reuse this dataset or integrate it with other data?YesAdditional CommentsIt does meet reuse criteria, but will be more reusable once the data is available from the Citrus Greening website.Any Additional Overall Comments to the AuthorRecommendationAccept

---

## [Reviewer Report]

Reviewer name and names of any other individual's who aided in reviewer Ruihan LiDo you understand and agree to our policy of having open and named reviews, and having your review included with the published papers. (If no, please inform the editor that you cannot review this manuscript.)YesIs the language of sufficient quality?YesPlease add additional comments on language quality to clarify if needed
Are all data available and do they match the descriptions in the paper? NoAdditional CommentsNo new sequencing data were generated in this paper. The author only modified the previous gene sets. Citation of raw data from the evidence supporting annotation mentioned in table2 should also be given. Are the data and metadata consistent with relevant minimum information or reporting standards? See GigaDB checklists for examples <a href="http://gigadb.org/site/guide" target="_blank">http://gigadb.org/site/guide</a>YesAdditional CommentsIs the data acquisition clear, complete and methodologically sound?NoAdditional CommentsIs there sufficient detail in the methods and data-processing steps to allow reproduction?NoAdditional CommentsThe method section of this paper seems too brief and should be supplemented in detailIs there sufficient data validation and statistical analyses of data quality? YesAdditional CommentsIs the validation suitable for this type of data?YesAdditional CommentsIs there sufficient information for others to reuse this dataset or integrate it with other data?YesAdditional CommentsAny Additional Overall Comments to the AuthorThe author of “Annotation of Putative Circadian Rhythm-Associated Genes in Diaphorina citri (Hemiptera : Liviidae)” manually annotated 27 genes related to circadian rhythm in the genome of Diaphorina citri. This study summarized the circadian model in D. citri which may help people to protect citrus from the citrus greening disease. But I have some comments that would need to be taken into consideration.

General comments:
The method section of this paper seems too brief and should be supplemented in detail according to these studied rhythm genes. 
No new sequencing data were generated in this paper. The author only modified the previous gene sets. Citation of raw data from the evidence supporting annotation mentioned in table2 should also be given. 

Specific comments:
Page 2: “…will allow future molecular therapeutics…” the molecular therapeutics for what? The object should be specified.
Page 3: The author only said "Based on the critical importance of the genes identified" in the section of Introduction, but did not introduce the circadian physiological habits of this insect, which like whether they stay in a tree all the time. Another problem is that the gene functions in table1 seem to be mostly related to development, reproduction and even death, but not directly related to circadian rhythms. The relationship between these genes with rhythms needs to be supplemented.
Table 1: Details of these references should be placed at the end of the article instead of in the table. Functional descriptions of these genes should be supplemented.
Page 6, “…, but also makes the D. melanogaster model different from non-dipterans due to their possession of cry2.” The D. Melanogaster model and the Drosophila Model are mentioned several times in this paragraph. Do they express the same meaning? Please use the same expression to avoid misunderstanding. 
Table 2：“X” and “space” should be explained in the table notes. In addition, the analysis methods of “MCOT, ISO-SEQ, RNA-Seq and Ortholog” in the "Evidence Supporting Annotation" are not described in this article. Are these results collected from an existing online database or generated from new analysis that author done in this study?
The section of Method is surprisingly simple. The necessary steps should be supplemented clearly. 
Figure2: If the author quote someone else's picture, please cite the sources of these references.
Figure 3 is meaningless and suggested to be removed.
Page12: How were “genome assembly errors” discovered? Did the author compare transcriptome data with genome data? Please supplement the method clearly.
RNA-Seq experimental analysis is mentioned in the conclusion, but there is no relevant experiment in this paper. Why included this sentence in the conclusion?RecommendationMajor Revision